# High-risk human papillomavirus genotypes among women of hill districts in Bangladesh

Ashrafun Nessa[1]*, Jannatul Ferdouse[1], Farhana Khatoon[1], Asma Akhter Sonia[1], Md Foyjul Islam[2], Lutfa Begum Lipi[3]

1 Department of Gynaecological Oncology, Bangladesh Medical University (BMU), Dhaka,Bangladesh, 2 Department of Epidemiology, Institute of Epidemiology, Disease Control and Research (IEDCR), Dhaka, Bangladesh, 3 Department of Obstetrics and Gynaecology, Dhaka Medical College Hospital (DMCH), Dhaka, Bangladesh

* ashrafun@bsmmu.edu.bd

## Abstract:

Knowledge on the distribution of type specific HPV (Human papillomavirus) genotypes in female cervix is crucial to identify women who are at a higher risk of developing cancer. This study aimed to find out the prevalence of High-risk HPV genotypes among women of three Hill districts of Bangladesh. This cross-sectional study was conducted between 1st January and 30th June 2024 among 1602 selected married asymptomatic tribal and nontribal women (30–60 years of age) at three Hill districts. Partial genotyping of HPV DNA specimens which detects the presence of 14 high-risk genotypes (including individual HPV-16, HPV-18, and others as a pooled group) was performed at Bangladesh Medical University (BMU). Women with previous treatment of cervical precancer and cancer, hysterectomy, cervical amputation and pregnancy were excluded. Statistical analysis utilized SPSS version 25.0, employing Chi-square and Fisher's Exact tests and P value <0.05 were considered significant. HR-HPV prevalence's were expressed as proportions. The influence of HR-HPV infection and socio-demographic factors was assessed using multinomial logistic regression analysis. The overall HR-HPV prevalence was 2.7% (n = 44) and 0.8% (n = 13) were tested positive for HPV16, 0.2% (n = 4) for HPV18 and 1.6% (n = 26) for 'Other HR-HPV' types. No significant difference of HR-HPV prevalence was observed among the three districts (p-value = 0.352) and among tribal (2.4%) and non-tribal/Bengali (3.2%) women (p-value>0.05). Higher number of marriages of the husbands have independent association with HR-HPV positivity showing an odds ratio of 2.02 (95% CI: 1.07–3.82, p = 0.030). The HR-HPV prevalence in hill districts of Bangladesh is low with independent association of higher number of marriages of the husbands with HR-HPV positivity. These findings may guide policymakers to initiate HR-HPV DNA-based screening and reconsider vaccination strategies in the hill areas, including the introduction of gender-neutral vaccination.

**Data availability statement:** To ensure participant privacy, the dataset is not publicly available but will be stored by the National Center for Cervical & Breast Cancer Screening & Training (NCCBCST). Data may be made available upon reasonable request to the corresponding authority via viacbe@bsmmu.edu.bd.

**Funding:** The study was financially supported by the MOHFW (Ministry of Health and Family Welfare, Bangladesh) of Bangladesh through the "Electronic Data Tracking with Population-based Cervical and Breast Cancer Screening Programme" (Code no. 16201-224259800). All the research activities, including were supported by he MOHFW's funding covered cost of participant enrolment, data and HPV sample collection, transport cost, essential laboratory expenses, etc ensuring the successful implementation of the study. The funders had no role in study design, data collection and analysis, decision to publish, or preparation of the manuscript.

**Competing interests:** The authors have declared that no competing interests exist.

## Introduction

Human papillomavirus (HPV) is a common sexually transmitted infection and the human anogenital tract can be infected by about 40 HPV types [1]. Based on the capacity to cause cervical cancer (CC), 15 HPVs are classified as high-risk (HR) and 12 as low-risk (LR) genotypes [2]. Studies showed that cervical dysplasia and cancer typically arises among women having persistent HR HPV infection [3,4]. Cervical Cancer is the fourth most common cancer in women worldwide, with approximately 660,000 new cases expected in 2022 and 94% of the 350,000 deaths resulting from CC occurred in low-and middle-income countries. The highest rates of both CC incidence and mortality were found in sub-Saharan Africa, Central America, and South-East Asia [5]. Cervical cancer is the second most frequently occurring cancer in Bangladeshi women, making up 13.3% of all cancers affecting females. In 2022, there were approximately 9,640 new cases of CC (11.6 per 100,000 women) and 5,826 deaths (7.0 per 100,000 women) [6–8]. The Government of Bangladesh (GOB) adopted visual inspection of acetic acid (VIA) method for CC screening free of cost for the women of 30–60 years of age and about 600 VIA centres exists at primary, secondary and tertiary health care facilities of 64 districts of Bangladesh [9–12]. High participation in screening programme and subsequent treatment of precancerous lesions can effectively reduced the CC related mortality [13]. GOB is continuing VIA-based screening programme. The low specificity of VIA and training of paramedical/ service providers are important concerns [14,15]. HPV-based screening is reliable, more effective and sensitive than cytology-based screening in preventing invasive CC, detecting persistent high-grade lesions earlier and provide a longer low-risk period [16]. Due to it's high negative predictive value (NPV), longer interval between screening visits reduces incidence and mortality of CC [17]. HPV detection became an acceptable option for the detection of cervical precancer necessitating treatment and HPV16 and HPV18 detection is more sensitive method and better choice than cytology-based cervical screening [18,19].

Previous studies in coastal districts reported HR-HPV prevalence around 2.6% among ever-married women, while broader regional surveys estimated HR-HPV burden between 0.5% and 7.1% across Bangladesh [20–22]. In mixed urban-rural cohorts, HR-HPV prevalence was approximately 4.2% [23]. However, no prior data exists for hill districts highlighting the significance of our study, which provides the first prevalence estimates in that setting. In Bangladesh the women of hill areas receive less health care and most of them live in hard-to-reach areas with limitations in access to health care services. Information on HR-HPV infection and its relation to different sociodemographic and reproductive factors have not been explored previously in hill areas. Such data will be useful in designing preventive strategies to reduce CC and related mortality in these areas. This study aimed to identify the type specific estimation of HR-HPV burden among a mixed tribal and Bengali population of hill areas to adopt better implementation strategies.

## Materials and methods

### Selection of study population

This cross-sectional study was conducted by the National Centre of Cervical and Breast Cancer Screening and Training (NCCBCST) at Bangladesh Medical University (BMU) between 1st January and 30th June 2024 among 1602 selected apparently healthy married women (30–60 years of age) of three Hill districts of Bangladesh (Bandarban, Khagrachori, Rangamati). Three upazilas were randomly selected by lottery from three districts and then again 3 community clinics were randomly selected by lottery from each upazila. (Fig 1)

Therefore, altogether nine community clinics were designated to collect samples. Each community clinic serves about 6000 populations at rural level with 600–800 women of target group for cervical cancer screening. Women frequently visit community clinic for essential primary health-care services including follow-up for themselves and their children. Outreach camps at designated community clinics were conducted to collect samples. Participants were recruited from women visiting the clinic for any reason and referred women by field staffs of related community clinics for screening. The target sample size for this study was approximately 1,800, calculated using a standard sample size estimation formula based on with a 95% confidence level, a precision of 1.2% and HPV prevalence of 7.1% among women without cervical abnormalities in Southern Asia [24]. However, the final recruited sample size was 1,602, due to time constraints and challenges in collecting samples from hard-to-reach areas. Sample size of each of the three selected districts were between 300 to 600.

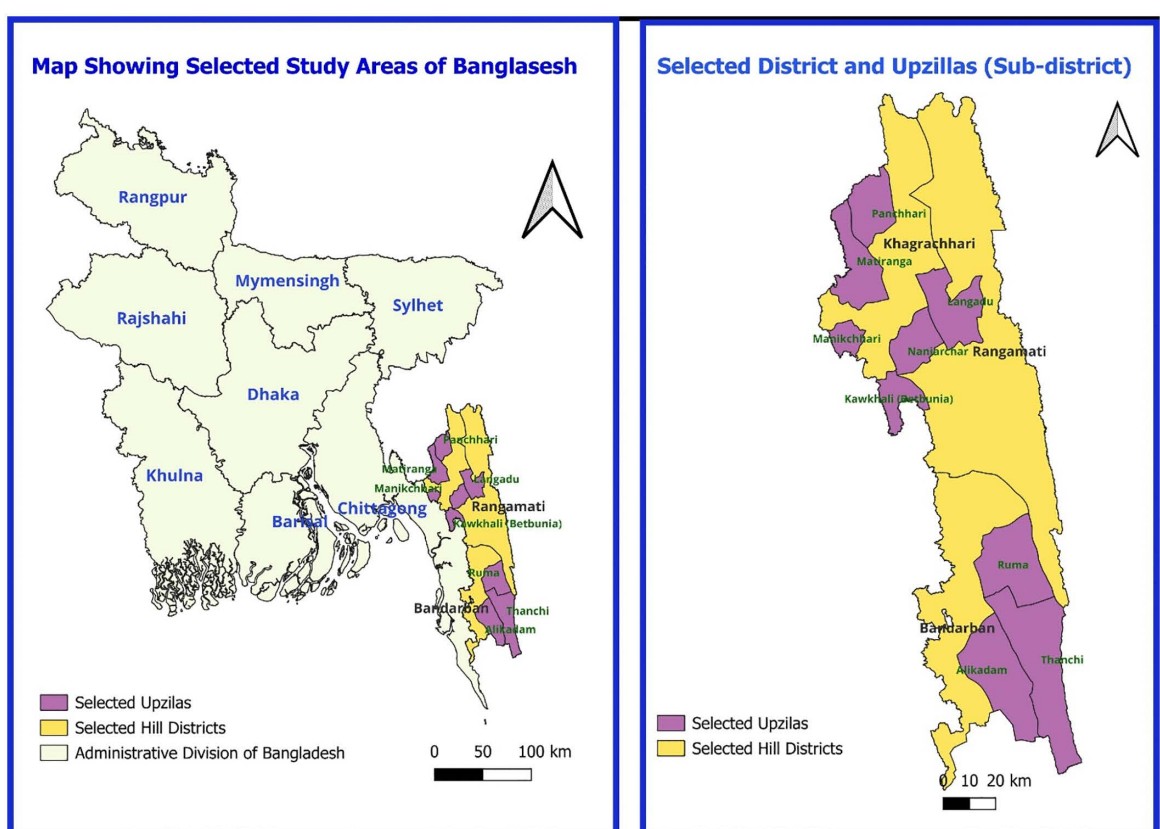

**Fig 1. Map showing selected three upazilas of three hill districts chosen for this study.** Map created using QGIS 3.34. Administrative boundaries for districts of Bangladesh provided by GADM, version 4.1, under a Creative Commons Attribution license (https://gadm.org).

Women with previous treatment due to cervical precancer and cancer, hysterectomy, cervical amputation and pregnancy were excluded. In total, 11 women were excluded for these reasons.

### Sample collection and transportation of specimen

The research team composed of research assistant and four VIA Trained SSNs/doctors. They were further trained on methodology of the study, counselling, motivation, recruitment of women, collection of samples for HPV test, placing of sample for HPV DNA testing in the standard transport medium (Cobas PCR collection media), interpretation of reports and referral of screen positive women. The data included women's socio-demographic and reproductive characteristics, VIA findings, HPV test report, colposcopy findings, histopathology report, etc. Women were counselled and written informed consent were collected. Tribal identity of participants was self-reported during enrollment. Efforts were made to recruit both tribal and non-tribal women proportionally across all three hill districts to ensure representative sampling of the study population. A gynaecological examination was performed along with introduction of Cuscos vaginal speculum. HPV DNA specimen were collected prior to VIA test using Cervical Sampler. Sample from cervical scrape was then suspended in a vial of preservative for transport to the laboratory. Specimens were transported to BMU within 14 days after collection. It does not require cold-chain management but should be stored at room temperature (<30°C).

### HPV DNA detection and genotyping

HR-HPV testing with partial genotype analysis was performed by a fully automated real-time PCR amplification and detection analyzer (Cobas 4800, Roche Diagnostics, GmbH, and Mannheim, Germany) to identify the presence of 14 high-risk genotypes at BMU. This test amplifies target DNA in cervical epithelial cells by polymerase chain reaction (PCR) and nucleic acid hybridization to allows specific identification of HPV types 16 and 18, and pooled detection of HPV types 31, 33, 35, 39, 45, 51, 52, 56, 58, 59, 66, and 68 [25]. The human beta-globulin-oriented fluorescent probe provided quality control of the whole reaction. Complete primer sequences, amplicon sizes, and reagent kit details are provided. [S1 Text]

### Ethical consideration

This research was conducted after obtaining Ethical clearance from the Institutional Review Board of BMU, Dhaka with registration number 989 and memo no (BSMMU/2023/12026). All participants were given informed written consent for participation.

### Statistical analysis

HR-HPV prevalences were expressed as proportions. Associations between HR-HPV infection and socio-demographic or reproductive factors were initially assessed using Chi-square or Fisher's Exact tests, as appropriate. Multinomial logistic regression was then performed to estimate odds ratios (ORs) and their 95% confidence intervals (CIs) for determinants of HR-HPV infection. Statistical significance was inferred at the 0.05 level. Only 44 HR-HPV-positive cases were available, resulting in a low number of events per variable (EPV) and a potential risk of overfitting. Model adequacy was assessed using the Hosmer–Lemeshow goodness-of-fit test and link test for specification, which indicated acceptable fit; however, the limited sample size may have reduced statistical power, and findings should be interpreted with caution. All analyses were conducted using SPSS version 25.0 (IBM Corp., Armonk, NY, USA) and Stata version 17.0 (StataCorp, College Station, TX, USA).

## Results

### HR-HPV prevalence and genotype distribution

Among the 1602 women in the hill areas, the overall HR-HPV prevalence was 2.7% (n = 44) and 0.8% (n = 13) were tested positive for HPV16, 0.2% (n = 4) for HPV18, 1.6% (n = 26) for Other HR-HPV types (Fig 2). Only 0.1% (n = 1) was co-infected with HPV16 and Other HR-HPV types.

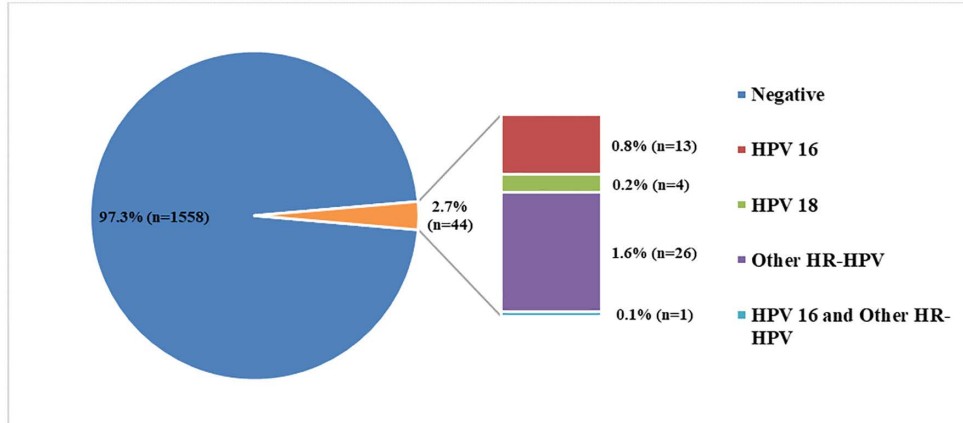

**Fig 2. HPV status of the study participant (n = 1602).**

Among the three hill districts, highest HR-HPV prevalence was found in Rangamati (3.7%, n = 15) followed by Bandarban (2.6%, n = 13) and Khagrachari (2.3%, n = 16). However, no significant difference of HR-HPV prevalence was observed among the three districts (p-value = 0.352). (Table 1). Among the 44 HPV-positive women, HPV 16 was the most common genotype (31.82%), followed by HPV 18 (9.09%), while the majority were infected with other high-risk HPV types (59.09%).(Table 2)

The HR-HPV positivity was lower (2.4%) among the tribal women than the non-tribal/Bengali (3.2%) women (Table 3). However, the difference of HR-HPV positivity among these two populations was not statistically significant (p-value>0.05).

## Age and HR-HPV prevalence and genotype distribution

Two rising trends of HR-HPV-positivity were observed between 30–39 age group (2.7% to 4.0%) and between 45–54 age group (2.0% to 6.0%) with a smaller peak in 35–39 years (4.0%, n = 15) and highest peak in 50–54 years (6.0%, n = 5). In the 55–60 age group, no HPV positive cases were observed. (Fig 3)

**Table 1. HR-HPV distribution among the three hill districts (n = 1602).**

| HPV | District | | | p-value |
|---|---|---|---|---|
| | Khagrachari | Rangamati | Bandarban | |
| | n (%) | n (%) | n (%) | |
| Negative | 685 (97.7) | 386 (96.3) | 487 (97.4) | 0.352$^{ns}$ |
| Positive | 16 (2.3) | 15 (3.7) | 13 (2.6) | |

*A Chi-square test was done ns=Non significant.

**Table 2. Genotypic Distribution of HR-HPV among HPV positive women (n = 44).**

| HPV Genotype | Frequency(n) | Percentage (%) |
|---|---|---|
| HPV 16 | 14 | 31.82 |
| HPV 18 | 4 | 9.09 |
| Other HR-HPV | 26 | 59.09 |
| Total | 44 | 100.00 |

**Table 3. HR-HPV prevalence among tribal and non-tribal population (n = 1602).**

| | HR-HPV Result | | *P-value |
|---|---|---|---|
| | Negative | | Positive |
| Tribal (n = 842) | 822 (97.6%) | 20 (2.4%) | 0.339[ns] |
| Non-tribal (n = 760) | 736 (96.8%) | 24 (3.2%) | |
| Total | 1558 (97.3%) | 44 (2.7%) | |

*Pearson Chi-square.

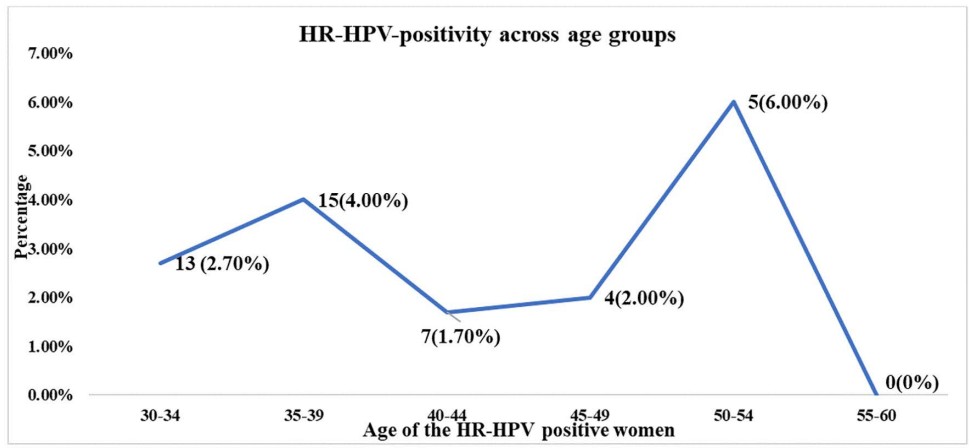

**Fig 3. Age-stratified HPV status among HR-HPV positive women (n = 44).**

Similarly, both HPV16 and HPV18 positivity showed two peaks in the 35–39 age group (38.5% and 50%) and 50–54 years age group (15.4% and 25%), although the second peak at 50–54 years is lower compared to the first peak. Other HR-HPV had the highest positivity in 30–34 years age group (34.6%) and positivity had a gradual decline by the age of 40 years (Fig 4).

## Association of HR-HPV DNA with socio-demographic and reproductive factors

The mean age of the study participants was 39.30 years (SD ± 6.91) and majority were between 30–44 years of age (78.9%). Only 15.9% of the women had secondary education and above. The majority were Jhum cultivators (29.4%), followed by laborers (22.4%), agricultural workers (16.0%), and housewives (14.0%).The majority of the women practiced Islam (48.5%) followed by Buddhism (38.4%). Half of the participants had monthly family income between 10,001–20,000 taka (50.6%). (Table 4)

Age, education, occupation, monthly income, number of marriage of women, age at marriage, age at first delivery, parity do not have any significant association with HR-HPV positivity. However, a significant association was found with the number of marriages of the husband, as 34.1% of HPV-positive individual's husbands had more than one marriage, showing an odds ratio of 2.02 (95% CI: 1.07–3.82, p = 0.030). (Table 5)

## Discussion

The overall prevalence of HR-HPV in the three hill districts is 2.7% and no significant difference of HR-HPV prevalence was observed among the three districts and between the tribal and non-tribal/Bengali women. The prevalence is lower

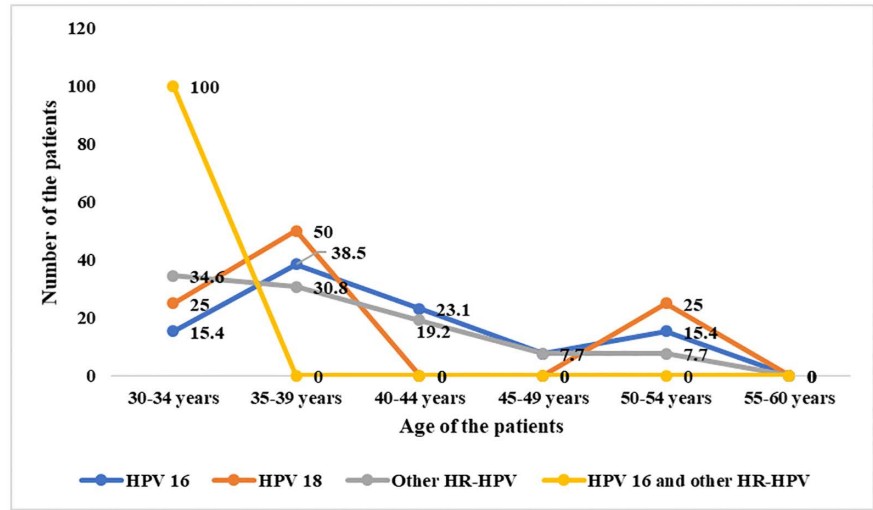

**Fig 4. Age-stratified HPV distribution HR-HPV-positive women (n = 44).**

than the reported HR-HPV prevalence in the general population of Bangladesh (3.6% and 4.2%) [20,23]. Furthermore, the prevalance is similar to the HR-HPV prevalence in the coastal districts of Bangladesh (2.6%) [21]. A hospital-based study in Chattrogram, southeastern part of Bangladesh, close to the hill tracts reported slightly higher (3.69%) prevalence of HPV genotypes 16 and 18 [26]. Women in the hilly areas of Maharashtra, India (8.7%), rural women in Nepal (14.4%), Bhutan (7.9%) have reported comparatively higher HR-HPV prevalence than the present study [27–29].Therefore, the HR-HPV prevalence among Bangladeshi women particularly in the hill and coastal areas are lower than the surrounding Asian countries. The prevalence is also lower than the worldwide (11.7%) and Southern Asian (7.1%) prevalence [24]. This low prevalence may be related to social and cultural factors, variation of sexual behaviour due to more conservative norms and religious traditions. The indigenous peoples have differed identities and engage largely in agriculture-based livelihood practices with a little diversification [30]. Moreover, a large number of hill population are Buddhist and Muslims and they have their own religious norms. The regional variation of HR-HPV prevalence is common, and related to different ages of the study populations, different methods of HPV detection, ongoing screening programme, variation in study designs and exposure to HPV risk factors [24].

There are differences of HPV prevalence among the tribal and non-tribal women population in India. A study in Karnataka, reported higher HPV prevalence among the tribal women (40.6%) than general population (14.3%), and in pre-adolescent tribal girls in Madhya Pradesh had double (6.6%) HPV prevalence, then that of urban girls of the same age group (3.3%) and these were related to sexual behaviour of tribal girls [31,32]. In contrast, this study in Bangladesh found lower HR-HPV prevalence among tribal women (2.4%) than in non-tribal women (3.2%). This may be related to social and cultural factors, similar sexual behaviour due to more conservative norms and religious traditions in Muslim and Buddhist communities.

The prevalence of HPV16 (0.8%) and HPV18 (0.2%) are close to a study in the central hilly region of Nepal (HPV 16–0.8%, HPV 18–2.3%) [28].The worldwide prevalence of HPV16 (3.2%) and HPV18 (1.4%) are higher than that of hill districts in Bangladesh [24]. However, in the hill regions of Northeast India with a high burden of CC, HPV16 (45%) and HPV18 (13%) positivity was alarmingly high [33]. Therefore, prevalence of HPV16 and HPV18 in the hill areas of Bangladesh are comparatively lower than many other countries and cancer control strategy in hill areas of Bangladesh can easily adopt HPV-based screening and treatment of all HPV16 and HPV18 positive women to prevent cervical cancer. As the

**Table 4. Sociodemographic and reproductive characteristics of the study population (n = 1602).**

| Variables | Frequency(n) | Percentage (%) |
|---|---|---|
| **Age (in years)** | | |
| 30-34 | 489 | 30.5 |
| 35-39 | 371 | 23.2 |
| 40-44 | 403 | 25.2 |
| 45-49 | 198 | 12.4 |
| 50-54 | 84 | 5.2 |
| 55-60 | 57 | 3.5 |
| **Mean±SD** | 39.30 ± 6.91 years | |
| **Education Level** | | |
| No formal education | 374 | 23.3 |
| Up to Primary | 974 | 60.8 |
| Secondary Level | 165 | 10.3 |
| Higher Secondary Level | 81 | 5.1 |
| Graduate and above | 8 | 0.5 |
| **Occupation** | | |
| Housewife | 225 | 14.0 |
| Service holder | 63 | 3.9 |
| Jhum cultivation | 471 | 29.4 |
| Labour | 359 | 22.4 |
| Agricultural work | 257 | 16.0 |
| Farming | 227 | 14.2 |
| **Religion** | | |
| Islam | 777 | 48.5 |
| Buddhism | 615 | 38.4 |
| Hinduism | 203 | 12.7 |
| Christianity | 7 | 0.5 |
| **Monthly family income (taka)** | | |
| Below 5,000 | 157 | 9.8 |
| 5,001-10,000 | 574 | 35.8 |
| 10,001-20,000 | 810 | 50.6 |
| 20,001-50,000 | 54 | 3.4 |
| Above 50,000 | 7 | 0.4 |

prevalence of HPV16 and HPV18 is low treatment of cervical precancer and HR-HPV positive women will be cost effective to reduce CC.

HR-HPV positivity showed an overall decline with increasing age, but an increase was observed for ages between 35–39 and 50–54. The latter group showing the highest HPV prevalence is mainly due to the small sample size for that age group, and this finding stresses the importance of more research on older age groups. A study on a semi-urban population in Bangladesh where samples were collected by self-sampling did not reflect this trend, with age groups older than 30–39 showing a decline in HR-HPV positivity [34].The HR-HPV-positivity in 35–39 age group in hill population indicates the importance of initiation of screening focusing this age group and larger implementation study should be arranged for this marginalized population.

The independent risk factor for positive cervical HR-HPV infection in this study is multiple marriages of husbands and 34.1% of HPV-positive individuals reported about husband's previous marriages (OR=2.02 (1.07–3.82). This

**Table 5. Sociodemographic and reproductive variables of the participants (N = 1602).**

| Variables | | Total | HPV | | OR (95% CI) | p-value |
|---|---|---|---|---|---|---|
| | | | Positive (n = 44) | Negative (n = 1558) | | |
| | | | | | | |
| Age (in years) | ≤45 years | 1367 (85.3) | 37 (84.1) | 1330 (85.4) | 1.10 (0.48-2.50) | 0.814[ns] |
| | >45 years | 235 (14.7) | 7 (15.9) | 228 (14.6) | | |
| Education of Women | Up to Primary | 1352 (84.3) | 35 (79.5) | 1316 (84.5) | 1.39 (0.66-2.94) | 0.376[ns] |
| | Secondary and above | 251 (15.7) | 9 (20.5) | 242 (15.5) | | |
| Occupation of Women | 288(17.98) | 10 (22.73) | 278 (17.84) | 288(17.98) | 0.738 (0.36-1.51) | 0.407[ns] |
| | 1314(82.02) | 34(77.27) | 1280 (82.16) | 1314(82.02) | | |
| Monthly Income (BDT) | ≤20,000 | 1541 (96.2) | 43 (97.7) | 1498 (96.1) | 1.72 (0.23-12.71) | >0.99[ns] |
| | >20,000 | 61 (3.8) | 1 (2.3) | 60 (3.9) | | |
| Number of marriages of women | One | 1584 (98.9) | 43 (97.7) | 1541 (98.9) | 2.10 (0.27-16.20) | 0.396[ns] |
| | Two or more | 18 (1.1) | 1 (2.3) | 17 (1.1) | | |
| Number of marriages of husband | One | 1270 (79.3) | 29 (65.9) | 1241 (79.7) | 2.02 (1.07-3.82) | **0.030\*** |
| | Two or more | 332 (20.7) | 15 (34.1) | 317 (20.3) | | |
| Age at marriage of women (years) | Up to 17 | 781 (48.8) | 23 (52.3) | 758 (48.7) | 1.15 (0.63-2.10) | 0.636ns |
| | 18 years and above | 821 (51.2) | 21 (47.7) | 800 (51.3) | | |
| Age at 1st delivery of women (years) | Up to 20 | 1386 (86.5) | 37 (84.1) | 1349 (86.6) | 1.22 (0.53-2.77) | 0.633ns |
| | Above 20 years | 216 (13.5) | 7 (15.9) | 209 (13.4) | | |
| Parity | Up to 2 | 867 (54.1) | 24 (54.5) | 843 (54.1) | 1.24 (0.38-4.07) | 0.954[ns] |
| | 3 and more | 735 (45.9) | 20 (45.5) | 715 (45.9) | | |

ns=Non-significant, *=significant.

suggests that husbands with multiple marriages have a higher chance of being HPV carriers, and introduction of Gender-neutral HPV vaccination program is a necessity. Several studies reported an association between HPV positivity and lifetime number of sexual partners [35,36]. While other sociodemographic and reproductive factors showed no significant association, this finding underscores the potential role of male partners in transmission; however, the limited number of positive cases in this study warrants cautious interpretation and validation in larger cohorts. The global pooled prevalence of HR-HPV was 21% with the predominance of HPV16 (5%, 95% CI 4–7) [37]. A significant number of males are therefore HPV-positive. Mathematical modeling has shown the application of a Gender-neutral vaccination is superior for eradication of oncogenic HPVs [38]. GOB introduced the HPV vaccine in the national program on 2 October 2023 and the bivalent vaccine roll-out has been planned to be implemented in all eight divisions of Bangladesh for 10–14 year old girls [39].

The three hill districts cover 10% of the total land areas of Bangladesh, but only 1% (1.842738) of the country's total population. About 11 different indigenous tribal ethnic groups constitutes 49.93% (N = 920217) of the population. Bengali population in these districts now represent 50.06% of the total hill districts population. Target Group of three Districts for cervical cancer screening is 312442 (30–60 years of age) [40]. GOB has established a comprehensive health service delivery system with enormous network of primary health care facilities from grassroots to higher levels throughout the country. However, marginalised and geographical dispersed communities and ethnic minorities, living in hard-to-reach areas frequently failed to avail the services [41]. Though 'VIA and CBE' facilities are available at the district hospitals and all UHCs, special outreach services need to be arranged at community clinics so that all women of the target population (both tribal and Bengali) can be covered by screening. HPV testing has high sensitivity, and the screening interval may be 10 years, making it highly suitable for use in hard-to-reach areas. Furthermore , HPV based screening and treatment is

proven cost-effective [42,43].Tribal population has a forest-based livelihood, language, cultural practice, religious faith and rituals [44]. They need special attention to motivate them, their family and tribal leaders need to be motivated and their culture and rituals need to be respected. Nurses and field workers of related health facilities should belong to both tribal and Bengali population. These marginalised populations with low incomes, have difficulties due to distances, transportation costs, travel time, and the necessity of accompaniment. All these factors need to be considered during developing strategy of screening for the hill tract population.

The sample size used and the remote nature of the hill tracts were limitations in this study. In Rangamati less samples could be collected due to nature of communication and political unrest. Only three upazilas of the hill areas may not be reflective of the entire area. A follow up study carried out on more people from more parts of the hill areas would provide a better impression on HPV prevalence, awareness of HPV and any socioeconomic factors of significance. Another limitation was younger and elderly group were not included. Detailed genotyping was not possible due to fund constrain.

## Conclusion

This study found a low prevalence (2.7%) of high-risk HPV among tribal and non-tribal women in the hill districts of Bangladesh, with HPV16 and HPV18 rates lower than global and regional averages. The significant association with husbands' multiple marriages underscores an important transmission pathway and the need for preventive measures. HPV-based screening offers a cost-effective alternative to VIA in hard-to-reach areas, requiring expansion through community-based outreach, culturally sensitive engagement, and improved diagnostic capacity. Alongside the current rollout for girls, gender-neutral HPV vaccination should be prioritized to reduce cervical cancer risk, promote equity, and support national and global elimination goals.

## Supporting information

**S1 Text. PCR primer targets and reagent kits Used in the cobas® HPV test.**
(PDF)

## Acknowledgments

We sincerely thank the health managers and healthcare providers at district, sub-district, and community clinics in the hill districts for their full cooperation during this study. The authors are also grateful to colleagues from the relevant departments of BMU for their support. Finally, we express our heartfelt appreciation to all participants who contributed to this research.

## Author contributions

**Conceptualization:** Ashrafun Nessa, Jannatul Ferdouse, Farhana Khatoon, Asma Akhter Sonia, Lutfa Begum Lipi, Md Foyjul Islam.

**Data curation:** Ashrafun Nessa, Jannatul Ferdouse, Lutfa Begum Lipi, Md Foyjul Islam.

**Formal analysis:** Ashrafun Nessa, Farhana Khatoon, Md Foyjul Islam.

**Funding acquisition:** Ashrafun Nessa.

**Investigation:** Ashrafun Nessa.

**Methodology:** Ashrafun Nessa, Jannatul Ferdouse, Farhana Khatoon, Asma Akhter Sonia, Lutfa Begum Lipi, Md Foyjul Islam.

**Project administration:** Ashrafun Nessa, Farhana Khatoon, Asma Akhter Sonia.

**Resources:** Ashrafun Nessa.

**Software:** Ashrafun Nessa, Md Foyjul Islam.

**Supervision:** Ashrafun Nessa, Jannatul Ferdouse, Farhana Khatoon, Asma Akhter Sonia, Lutfa Begum Lipi, Md Foyjul Islam.

**Validation:** Ashrafun Nessa, Asma Akhter Sonia, Lutfa Begum Lipi, Md Foyjul Islam.

**Writing – original draft:** Ashrafun Nessa, Jannatul Ferdouse, Farhana Khatoon, Asma Akhter Sonia, Lutfa Begum Lipi, Md Foyjul Islam.

**Writing – review & editing:** Ashrafun Nessa, Jannatul Ferdouse, Farhana Khatoon, Asma Akhter Sonia, Lutfa Begum Lipi, Md Foyjul Islam.

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
