## [Decision Letter · Decision Letter 0]

6 Aug 2025

Dear Dr. Nessa,

Thank you for submitting your manuscript to PLOS ONE. After careful consideration, we feel that it has merit but does not fully meet PLOS ONE’s publication criteria as it currently stands. Therefore, we invite you to submit a revised version of the manuscript that addresses the points raised during the review process.

We look forward to receiving your revised manuscript.

Kind regards,

Anoop Kumar, Ph.D.

Academic Editor

PLOS ONE

Journal Requirements:

1. Please ensure that your manuscript meets PLOS ONE's style requirements, including those for file naming. The PLOS ONE style templates can be found at https://journals.plos.org/plosone/s/file?id=wjVg/PLOSOne_formatting_sample_main_body.pdf and https://journals.plos.org/plosone/s/file?id=ba62/PLOSOne_formatting_sample_title_authors_affiliations.pdf.

- Chakraborty S, Nessa A, Ferdous NE, Rahman MM, Rashid MHU, Sonia AA, Islam MF. Prevalence and genotypic distribution of high-risk human papillomavirus (HPV) among ever-married women in coastal regions of Bangladesh. PLoS One. 2024 Dec 12;19(12):e0313396. doi: 10.1371/journal.pone.0313396. PMID: 39666707; PMCID: PMC11637305.

In your revision ensure you cite all your sources (including your own works), and quote or rephrase any duplicated text outside the methods section. Further consideration is dependent on these concerns being addressed.

 [The study was financially supported by the MOHFW of Bangladesh through the “Electronic Data Tracking with Population-based Cervical and Breast Cancer Screening Programme” (Code no. 16201-224259800). All the research activities, including were supported by he MOHFW’s funding covered cost of participant enrolment, data and HPV sample collection, transport cost, essential laboratory expenses, etc  ensuring the successful implementation of the study.].

4. Please expand the acronym “MOHFW” (as indicated in your financial disclosure) so that it states the name of your funders in full.

[We pay our respect to the technical and financial support from of the Ministry of Health and Family Welfare (MOHFW) of Bangladesh, Special thanks are given to the health managers and healthcare providers at healthcare facilities at the district, sub-district, community clinics of the hill districts for their full cooperation. The authors are thankful to the colleagues of relevant departments of BMU. The authors would also like to express their gratitude to the participants of this study.]

 [The study was financially supported by the MOHFW of Bangladesh through the “Electronic Data Tracking with Population-based Cervical and Breast Cancer Screening Programme” (Code no. 16201-224259800). All the research activities, including were supported by he MOHFW’s funding covered cost of participant enrolment, data and HPV sample collection, transport cost, essential laboratory expenses, etc  ensuring the successful implementation of the study.]

6. In the online submission form, you indicated that [Most of the data collected were analyzed and are contained within this published article. To maintain data privacy, the data used are not publicly available. The data can be made available after a rational request from the researchers.].

Reviewers' comments:

Reviewer's Responses to Questions

**Comments to the Author**

1. Is the manuscript technically sound, and do the data support the conclusions?

Reviewer #1: Partly

Reviewer #2: Yes

2. Has the statistical analysis been performed appropriately and rigorously?

Reviewer #1: Yes

Reviewer #2: Yes

3. Have the authors made all data underlying the findings in their manuscript fully available?

Reviewer #1: No

Reviewer #2: Yes

4. Is the manuscript presented in an intelligible fashion and written in standard English?

Reviewer #1: No

Reviewer #2: Yes

Reviewer #1: The study explores the prevalence and distribution of high-risk human papillomavirus (HR-HPV) genotypes among married tribal and non-tribal women aged 30–60 years in three hill districts of Bangladesh. Using PCR-based partial genotyping (via the Cobas 4800 system), it examines associations between HR-HPV infection and various sociodemographic factors. The research tackles an important public health concern in a geographically isolated and underrepresented population. Overall, the study design, data collection methods, and statistical approach are appropriate for a cross-sectional analysis.

However, several issues affect the manuscript’s clarity, structure, presentation of data, and depth of interpretation:

1. Language in the abstract needs polishing-the phrase “These findings direct policy makers” is awkward and should be rephrased for smoother readability.

2. The term “partial genotyping” should either be briefly explained in the abstract or removed, especially if space is limited.

3. The authors should clearly specify that only HPV16 and HPV18 were genotyped individually, while the remaining types were detected as a pooled group.

4. In the introduction, the research gap is not well justified. The manuscript assumes a lack of prior HPV data in hill districts without providing supporting evidence or references.

5. In the methods section, the sample size calculation is unclear. The assumed HPV prevalence of 7.1% appears arbitrary, and no information is provided about statistical power or margin of error.

6. The statistical analysis section contains repetitive descriptions and should be made more concise and precise.

7. The authors should clarify whether tribal and non-tribal women were proportionally represented across all three districts and explain how tribal identity was determined; was it self-reported?

8. In the results section, Figures 3 and 4 lack clarity—axis labels and legends need improvement for better readability. Additionally, the low number of HPV-positive cases (only 4 cases of HPV18) weakens the validity of subgroup analysis.

9. Statistical concerns exist regarding the logistic regression analysis. With only 44 HR-HPV-positive cases, the model may be underpowered. It is unclear whether diagnostic checks or goodness-of-fit tests were performed.

10. In the discussion, assertions such as “screening and treatment will be cost-effective” need to be supported by data or citations.

11. The manuscript contains several grammatical and typographical errors, such as using “leave in hard-to-reach areas” instead of “live” and unclear phrases like “number of marriages of the husbands have independent association.” A thorough language review by a native or fluent English speaker is recommended.

12. The data availability statement is insufficient. PLOS ONE requires open and accessible data, and the phrase “data available on request” does not meet their standards.

Reviewer #2: In the manuscript titled “High-risk Human Papillomavirus Genotypes among Women of Hill Districts in Bangladesh,” the authors have conducted a well-designed study that addresses several important research questions. Following minor revisions, the manuscript is suitable for publication. The study is thorough, methodologically sound, and contributes valuable insights to the existing body of knowledge.

Comments to Authors:

1. Nothing has been mentioned regarding taking a consent from the cohort of 1602 selected apparently healthy married women (30-60years) for the HPV tests. Were they informed that a study was being created and it would be a great addition to have a draft consent form in the paper as supplementary material.

2. It would also be great to know what percentage of the people were excluded due to cervical precancer or anyone who came in knowing that they were HPV positive. I would be very curious to know, if anyone was aware of it at all.

3. Could you please provide all the scientific details regarding the PCR primers used during the experiment in the supplementary to provide more transparency and reproducibility in the scientific world.

4. It would also be nice to create a small table regarding the different types of HPV like HPV 16, 12, 18 and others.

5. The conclusion can be made a bit stronger with these results, at present the conclusion feels like a future direction of two sentences.

**Do you want your identity to be public for this peer review?** For information about this choice, including consent withdrawal, please see our Privacy Policy

Reviewer #1: No

Reviewer #2: No

---

## [Decision Letter · Decision Letter 1]

18 Nov 2025

High-risk Human Papillomavirus Genotypes among Women of Hill Districts in Bangladesh

PONE-D-25-31719R1

Dear Dr. Ashrafun Nessa,

We’re pleased to inform you that your revised manuscript has been judged scientifically suitable for publication and will be formally accepted for publication once it meets all outstanding technical requirements.

Kind regards,

Dr. Minal Dakhave

Academic Editor

PLOS ONE

Additional Editor Comments (optional):

Reviewers' comments:

Reviewer's Responses to Questions

**Comments to the Author**

Reviewer #2: All comments have been addressed

2. Is the manuscript technically sound, and do the data support the conclusions?

Reviewer #2: Yes

3. Has the statistical analysis been performed appropriately and rigorously?

Reviewer #2: Yes

4. Have the authors made all data underlying the findings in their manuscript fully available?

Reviewer #2: Yes

5. Is the manuscript presented in an intelligible fashion and written in standard English?

Reviewer #2: Yes

Reviewer #2: (No Response)

**Do you want your identity to be public for this peer review?** For information about this choice, including consent withdrawal, please see our Privacy Policy

Reviewer #2: **Yes: ** Amrita Basu

---

## [Editor Report · Acceptance letter]

PONE-D-25-31719R1

PLOS ONE

Dear Dr. Nessa,

I'm pleased to inform you that your manuscript has been deemed suitable for publication in PLOS ONE. Congratulations! Your manuscript is now being handed over to our production team.

Kind regards,

on behalf of

Dr. Minal Dakhave

Academic Editor

PLOS ONE